# Expanding SPHERE-JEPA: A Family of Statistical Regularizers for the Hypersphere

## Abstract

In Self-Supervised Learning (SSL), preventing representation collapse by explicitly enforcing a uniform distribution on the unit hypersphere has proven to be effective. However, current frameworks typically rely on *sliced* statistical regularizers such as SIGReg (used in LeJEPA) and SUSReg (used in SPHERE-JEPA), which approximate this continuous objective via Monte Carlo sampling along random 1D directions. This stochasticity injects projection variance into the training gradients, destabilizing optimization, and hindering convergence. In this work, we first show that analytically integrating out these random projections natively yields a deterministic Maximum Mean Discrepancy (MMD), bypassing the variance of sliced methods. Motivated by this equivalence, we formulate full-dimensional objectives for MMD, Kernel Stein Discrepancy (KSD), and Kullback–Leibler (KL) divergence directly on the sphere to enforce a uniform distribution. To prevent spatial bias, we equip these tests with rotationally invariant kernels constructed via spectral theory, systematically evaluating two canonical families: smooth exponential decay (Heat) and strict frequency cutoff (Bandlimited) filters. Empirically, removing projection-induced noise results in more stable optimization, faster convergence, and consistent improvements over stochastic sliced regularizers on ImageNet and Galaxy10. Furthermore, we reveal that the choice of the statistical test shapes the geometry of the learned latent space: MMD and KSD favor locally clustered organization suitable for object-centric domains, whereas the continuous KDE-based KL divergence promotes fine-grained instance separation, yielding the strongest results on unclustered procedural texture retrieval.

## 1 Introduction

A central paradigm in modern Self-Supervised Learning (SSL) is the multi-view formulation, where an encoder learns representations by minimizing an invariance objective across augmented views of the same input (Chen et al., 2020; Grill et al., 2020). To prevent representation collapse, modern architectures employ explicit global regularization (Balestriero and LeCun, 2025). Recent findings demonstrate that enforcing a uniform distribution on the unit hypersphere $\mathbb{S}^{d-1}$ is an optimal target, minimizing worst-case errors for downstream evaluations (Nicollier et al., 2026).

To enforce hyperspherical uniformity, frameworks such as SPHERE-JEPA (Nicollier et al., 2026) (via SUS-Reg) rely on *sliced* statistical regularizers, which project representations onto random 1D directions and compute the Epps-Pulley (Epps and Pulley, 1983) (EP) test. However, relying on this Monte Carlo approximation naturally injects "projection variance" into the training gradients, destabilizing optimization and slowing convergence. Concurrently, KerJEPA (Zimmermann et al., 2025) demonstrated that analogous sliced regularizers enforcing Gaussian embeddings can be analytically integrated to entirely bypass this variance, resulting in stable, closed-form Maximum Mean Discrepancy (MMD).

In this work, we extend this insight to the uniform distribution on the sphere. Building upon the equivalence established by KerJEPA, we demonstrate that analytically integrating out the random projections over $\mathbb{S}^{d-1}$ similarly yields a deterministic, closed-form Maximum Mean Discrepancy (MMD). Motivated by the strict suboptimality of 1D proxies, we focus only on exact multivariate statistics. While KerJEPA successfully

employed MMD and Kernel Stein Discrepancy (KSD) (Liu et al., 2016) to enforce unconstrained Gaussian priors in Euclidean spaces, we adapt these exact multivariate statistical tests—alongside the Kullback-Leibler (KL) divergence (Kullback and Leibler, 1951)—to operate directly on the manifold, thereby enforcing hyperspherical uniformity.

Because standard empirical distribution tests do not generalize beyond 1D, exact high-dimensional statistical testing intrinsically relies on kernel methods. However, to ensure that the kernel does not privilege any specific region of the hypersphere $\mathbb{S}^{d-1}$, we restrict our focus to rotationally invariant kernels. Any such kernel is characterized by the Laplacian's eigenvalues. Leveraging this spectral foundation, we systematically evaluate three distinct kernels: the exact kernel implicitly induced by the analytical integration of SUSReg (Nicollier et al., 2026), alongside two canonical frequency filters adapted to the sphere: the Heat kernel (acting as a smooth exponential decay) and the Bandlimited kernel (enforcing a strict frequency cutoff).

In summary, our main contributions are as follows:

- **Deterministic Hyperspherical Regularization:** We derive a deterministic formulation of the SUSReg objective by analytically integrating the random 1D projections. This results in a closed-form Maximum Mean Discrepancy (MMD) objective directly defined on $\mathbb{S}^{d-1}$, removing Monte Carlo projection sampling during training.

- **Hyperspherical Statistical Regularizers:** We adapt multivariate statistical objectives—Maximum Mean Discrepancy (MMD), Kernel Stein Discrepancy (KSD), and Kullback–Leibler (KL) divergence—to enforce uniform priors directly on the hyperspherical manifold.

- **Kernel Choice from Spectral Characterization:** Guided by the spectral characterization of rotationally invariant kernels on the sphere, we study two kernel families on the hypersphere: Heat and Bandlimited kernels.

- **Improved Representation Quality:** Across MMD and KSD variants, our deterministic objectives consistently outperform SUSReg, yielding gains of up to $+1.3/+1.6$ points (linear/$k$-NN) on ImageNet-100 and $+5.1/+4.1$ points on Galaxy10.

The remainder of this paper is organized as follows. Section 2 reviews the standard multi-view SSL framework on the hypersphere. In Section 3, we demonstrate the suboptimality of sliced proxies and establish their equivalence to a closed-form MMD. Section 4 introduces the exact multivariate statistical tests, and Section 5 details the design of our rotationally invariant spectral kernels. We derive explicit closed-form training objectives in Section 6. Finally, Section 7 presents our empirical evaluation across multiple benchmarks, and Section 8 concludes the work. Detailed proofs and derivations are deferred to the Appendices.

## 2 Preliminaries: Self-Supervised Learning on the Hypersphere

A central paradigm in modern Self-Supervised Learning (SSL) is the multi-view formulation, in which different augmented views of the same sample are used as supervisory signals (Chen et al., 2020; Grill et al., 2020). Formally, we consider a dataset of $N$ independent samples. Each sample is processed through a stochastic data augmentation pipeline to generate $V_a$ views $x_{n,v} \in \mathbb{R}^D$, where $n \in \{1, \ldots, N\}$ and $v \in \{1, \ldots, V_a\}$. Among these, we distinguish $V_g \leq V_a$ *global* views—which typically correspond to large-scale crops—from the remaining *local* views.

An encoder network $f_\theta$ maps each view to a representation $z_{n,v} = f_\theta(x_{n,v}) \in \mathbb{R}^d$. Following the exact formulation introduced by SPHERE-JEPA (Nicollier et al., 2026), these embeddings are subsequently $\ell_2$-normalized to lie on the unit hypersphere:

$$\tilde{z}_{n,v} = \frac{z_{n,v}}{\|z_{n,v}\|} \in \mathbb{S}^{d-1}.$$

To enforce representation consistency across augmentations, the model minimizes an alignment objective. This multi-view invariance loss can be formulated as:

$$\mathcal{L}_{\text{inv}} = \frac{1}{V_a} \sum_{v=1}^{V_a} \|\tilde{z}_{n,v} - \mu_n\|_2^2, \qquad \mu_n = \frac{1}{V_g} \sum_{v'=1}^{V_g} \tilde{z}_{n,v'}. \tag{1}$$

However, optimizing $\mathcal{L}_{\text{inv}}$ alone is degenerate, as the network can easily achieve zero loss by mapping all inputs to a single constant vector (Grill et al., 2020) (representation collapse). To prevent such trivial solutions and strictly control the global geometry of the representation space, SSL frameworks introduce an explicit regularization term (Caron et al., 2021; Balestriero and LeCun, 2025), leading to the general training objective:

$$\mathcal{L} = (1 - \lambda)\mathcal{L}_{\text{inv}} + \lambda\mathcal{L}_{\text{reg}}, \tag{2}$$

where $\lambda \in (0, 1)$ is a hyperparameter that controls the trade-off between the two objectives. In this formulation, while $\mathcal{L}_{\text{inv}}$ attracts views of the same instance, $\mathcal{L}_{\text{reg}}$ constrains the global distribution of the embeddings. Based on spherical uniformity optimality (Nicollier et al., 2026), our goal is to design $\mathcal{L}_{\text{reg}}$ such that it explicitly enforces the uniform distribution on the hypersphere, $q = \text{Unif}(\mathbb{S}^{d-1})$. The critical question then becomes: how should we practically compute the discrepancy between the empirical embeddings and this continuous uniform target?

## 3 Beyond Sliced Methods: Full-Dimensional Tests

To regularize high-dimensional representation spaces, recent frameworks (Balestriero and LeCun, 2025; Nicollier et al., 2026) rely on *sliced* methods. Justified by the Cramér-Wold theorem (Cuesta-Albertos et al., 2007), these approaches project the embeddings onto random 1D directions, effectively reducing the complex multi-dimensional distribution matching problem into a series of tractable 1D statistical tests. However, approximating this continuous objective via Monte Carlo sampling inherently injects an artificial projection variance into the training gradients. Fortunately, this stochasticity is unnecessary.

**Equivalence to a Closed-Form MMD.** By leveraging Bochner's (Rahimi and Recht, 2007) and Fubini's theorems, we can analytically integrate out all possible random projection directions $a \in \mathbb{S}^{d-1}$. As detailed in Appendix A, we apply the analytical integration framework from KerJEPA (Zimmermann et al., 2025)—which has recently established this for the Gaussian distribution—to the uniform distribution on $\mathbb{S}^{d-1}$. Specifically, let $X$ and $Y$ be random variables on $\mathbb{S}^{d-1}$ (representing, for instance, the empirical embedding distribution and the uniform target), and let $\text{EP}(\cdot, \cdot)$ denote the 1D Epps-Pulley discrepancy (Balestriero and LeCun, 2025). This exact integration reveals that the expected sliced Epps-Pulley discrepancy natively yields a deterministic Maximum Mean Discrepancy (Gretton et al., 2012) (MMD) with an induced kernel $\bar{k}$:

$$\mathbb{E}_{a \sim \text{Unif}(\mathbb{S}^{d-1})}\left[\text{EP}(a^\top X, a^\top Y)\right] = \text{MMD}_{\bar{k}}^2(X, Y). \tag{3}$$

Crucially, because the projection directions are distributed uniformly, the induced kernel $\bar{k}$ is rotationally invariant and admits the following explicit 1D integral representation:

$$\bar{k}(x, y) = \int_{-1}^{1} \exp\left(-(1 - x^\top y)t^2\right) \rho_d(t)\, dt, \tag{4}$$

where $\rho_d(t) \propto (1 - t^2)^{\frac{d-3}{2}}$ is the marginal density of the hypersphere. In practice, we compute this exact kernel using Gauss-Jacobi quadrature. This numerical scheme naturally absorbs the geometric density $\rho_d(t)$ into its precomputed weights, transforming the analytical integral into a simple, deterministic dot product.

**The Cost of Projection Variance.** Compared to our exact formulation, evaluating the sliced objective via standard Monte Carlo sampling inherently suffers from projection noise (Figure 1). This artificial variance destabilizes the training gradients and bounds optimization efficiency, converging to the true MMD only in the infinite-projection limit. By eliminating this stochasticity, our deterministic closed-form MMD achieves notably faster convergence and superior training dynamics, as demonstrated by the accelerated training

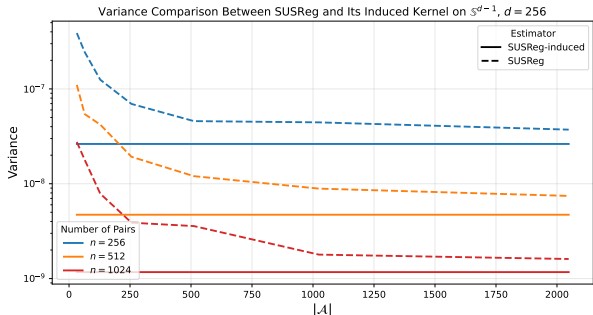
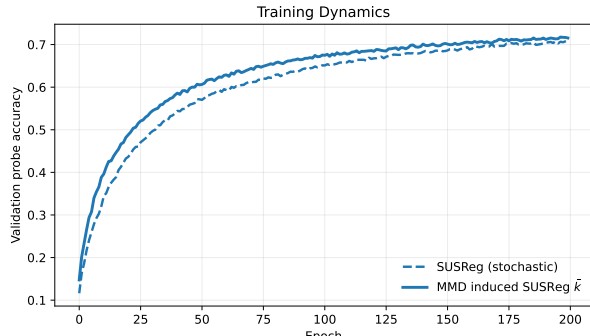

Figure 1: **Projection Variance.** Variance comparison between standard SUSReg (dashed) and its induced MMD estimator (solid) on $\mathbb{S}^{255}$.

Figure 2: **Training Dynamics.** The closed-form MMD (solid) entirely eliminates projection noise, achieving notably faster convergence.

convergence on ImageNet-100 (Figure 2). Motivated by the strict empirical and theoretical suboptimality of 1D proxies, we discard sliced approximations to focus on exact multivariate discrepancies natively on the manifold.

## 4 Statistical Tests for Hyperspherical Uniformity

As established in Section 2, our objective is to match the empirical distribution of representations with the uniform target $q = \mathrm{Unif}(\mathbb{S}^{d-1})$. For each view $v$, let $\hat{p}_v$ denote the empirical distribution of normalized embeddings $\{\tilde{z}_{n,v}\}_{n=1}^{N}$. To enforce uniformity across views, we minimize the general regularization loss

$$\mathcal{L}_{\mathrm{reg}} = \frac{1}{V_a} \sum_{v=1}^{V_a} D(\hat{p}_v, q), \tag{5}$$

where $D$ is a statistical discrepancy measure between probability distributions. The choice of this discrepancy is not a detail; it dictates how the algorithm behaves and shapes the geometry of the learned latent space. Because different statistical tests penalize deviations from uniformity in distinct ways, they implicitly favor different topological arrangements—such as locally clustered spaces suited for object-centric classification, or unclustered, mutually repelled spaces optimal for continuous textures.

For notational simplicity, we drop the view index $v$ when the context is clear, denoting the empirical distribution simply as $\hat{p}$.

Motivated by the exact high-dimensional equivalence demonstrated in Section 3, we bypass stochastic 1D approximations to evaluate $D(\hat{p}, q)$ directly through full-dimensional statistical tests on the manifold. All the following discrepancies rely on a positive definite base kernel $k : \mathbb{S}^{d-1} \times \mathbb{S}^{d-1} \to \mathbb{R}$. We introduce and formulate objectives for the following three distinct statistical tests:

**Maximum Mean Discrepancy (MMD).** MMD quantifies the distance between distributions by comparing their kernel mean embeddings (Gretton et al., 2012). In our framework, the squared MMD between the empirical batch distribution $\hat{p}$ and the continuous uniform target $q$, given a positive definite base kernel $k$, expands as:

$$D_{\mathrm{MMD}^2}(\hat{p}, q) = \mathbb{E}_{x,x'\sim\hat{p}}[k(x,x')] + \mathbb{E}_{y,y'\sim q}[k(y,y')] - 2\mathbb{E}_{x\sim\hat{p},y\sim q}[k(x,y)]. \tag{6}$$

As detailed in Appendix C, because the target $q$ is uniform and the kernel is rotationally invariant, all expectations with respect to $q$ reduce to analytic constants. As a result, the training objective $D_{\mathrm{MMD}}(\hat{p}, q)$ simplifies: it relies solely on pairwise kernel similarities within the empirical batch $\hat{p}$, eliminating the need for target sampling.

**Kernel Stein Discrepancy (KSD).** KSD quantifies distribution mismatch by evaluating how well empirical samples satisfy Stein's identity for a target distribution (Liu et al., 2016). Formally, the squared KSD for our empirical batch $\hat{p}$ against the uniform target $q$ is defined as the expectation of a Stein kernel $k_q$:

$$D_{\text{KSD}^2}(\hat{p}, q) = \mathbb{E}_{x,x' \sim \hat{p}}[k_q(x, x')]. \tag{7}$$

The full construction of this Stein kernel, detailed in Appendix B, relies on the target distribution's score function, $s_q(x) = \nabla_x \log q(x)$, and a base reproducing kernel $k$. Because the uniform target $q$ has a constant density on the hypersphere $\mathbb{S}^{d-1}$, its score function vanishes. Consequently, the objective $D_{\text{KSD}}(\hat{p}, q)$ can be evaluated in closed form using only the base kernel's first and second derivatives with respect to the pairwise cosine similarity.

**Kullback-Leibler (KL) Divergence via Kernel Density Estimation.** The standard KL divergence measures the relative entropy between distributions. To match our empirical batch $\hat{p}$ with the uniform target $q$, the objective takes the form:

$$\text{KL}(\hat{p}\|q) = \mathbb{E}_{x \sim \hat{p}}[\log \hat{p}(x) - \log q(x)]. \tag{8}$$

However, this standard formulation cannot directly compare the discrete empirical distribution $\hat{p}$ against the continuous target $q$. To resolve this domain gap, we construct a continuous surrogate for the batch using a Kernel Density Estimator (KDE), $\tilde{p}(x) = \mathbb{E}_{x' \sim \hat{p}}[k(x, x')]$. Assuming this smooth surrogate faithfully captures the underlying data distribution, substituting it into the equation above yields our tractable sample-based objective:

$$D_{\text{KL}}(\hat{p}, q) \approx \mathbb{E}_{x \sim \hat{p}}[\log \tilde{p}(x) - \log q(x)]. \tag{9}$$

## 5 Spectral Kernel Design on the Hypersphere

The effectiveness of MMD, KSD, and KL-based discrepancies depends on the properties of the kernel $k$. To prevent the regularization objective from biasing the representation space toward any preferred direction, we constrain the kernel to be rotationally invariant (zonal): $k(x, y) = \varphi(x^\top y)$. We systematically normalize it so that $\varphi(1) = 1$.

Viewed through the lens of spectral theory, the hypersphere is equipped with the Laplace-Beltrami operator, whose eigenfunctions are the spherical harmonics. By Schoenberg's theorem (Schoenberg, 1942), any valid positive-definite zonal kernel on $\mathbb{S}^{d-1}$ admits a spectral expansion over normalized Gegenbauer polynomials $\tilde{C}_\ell^{(\alpha)}$ with non-negative coefficients. Following the constructions of Giné and Wahba (Giné, 1975; Wahba, 1981), we can explicitly control the smoothness of the Reproducing Kernel Hilbert Space (RKHS) in which we measure the discrepancies by defining these coefficients as weights $w(\lambda_\ell)$ that depend on the smoothness of the corresponding mode:

$$\varphi(c) = \frac{1}{Z} \sum_{\ell=0}^{\infty} w(\lambda_\ell) \, \tilde{C}_\ell^{(\alpha)}(c), \tag{10}$$

where $c = x^\top y$, $\alpha = (d-2)/2$, and $\lambda_\ell = \ell(\ell + d - 2)$ are the Laplacian eigenvalues. The normalization constant $Z = \sum_{\ell=0}^{\infty} w(\lambda_\ell) \, \tilde{C}_\ell^{(\alpha)}(1)$ enforces $\varphi(1) = 1$.

Importantly, each eigenvalue $\lambda_\ell$ equals the Dirichlet energy of its corresponding eigenfunction. Because these kernels heavily penalize high-energy (unsmooth) functions, statistical tests derived from this general expansion are broadly referred to as "Sobolev tests" in the statistics literature (Giné, 1975; Jupp, 2008).

In practice, the spectral weights $w(\lambda_\ell)$ act as frequency filters. Low values of $\ell$ correspond to smooth, global geometric variations, while higher values capture increasingly oscillatory spatial patterns. This framework naturally motivates two canonical filter choices natively adapted to the sphere's geometry (illustrated in Figure 3):

- **Heat Kernel (Smooth Decay):** Applying an exponential decay $w(\lambda_\ell) = e^{-t\lambda_\ell}$, where $t > 0$ is a scale parameter, yields the heat kernel (Zhao and Song, 2018). This progressive damping of highly oscillatory patterns results in a stable, multi-scale similarity measure.

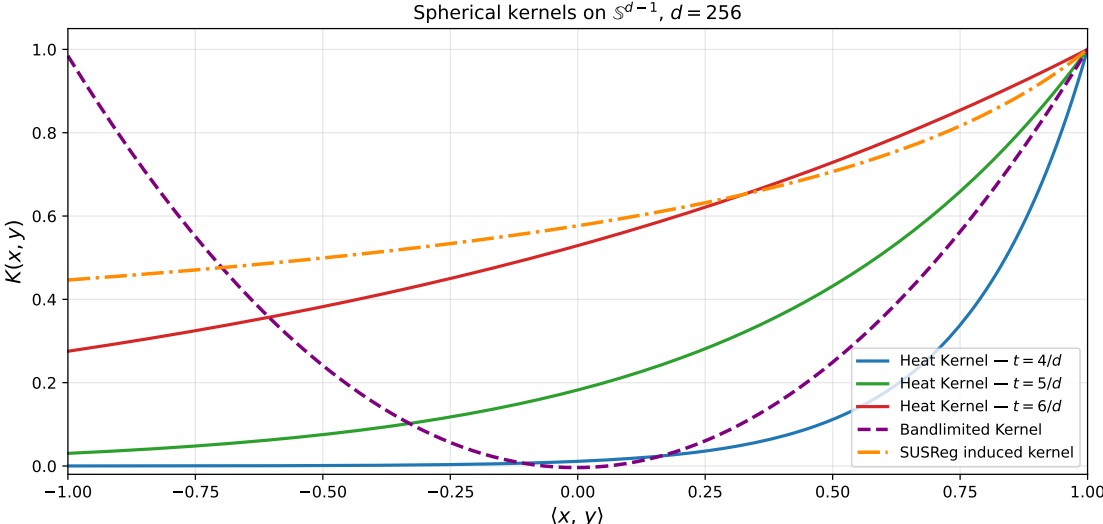

Figure 3: **Profiles of normalized zonal kernels on the hypersphere** $\mathbb{S}^{d-1}$ **($d = 256$).** We compare the heat kernel at different scale parameters ($t \in \{4/d, 5/d, 6/d\}$) and the Bandlimited kernel against the implicit kernel induced by the SUSReg objective. All kernels are evaluated as a function of the pairwise cosine similarity $c = x^\top y \in [-1, 1]$.

- **Bandlimited Kernel (Hard Cutoff):** Alternatively, applying a strict low-pass filter $w(\lambda_\ell) = \mathbf{1}_{\ell \leq L}$ yields a bandlimited kernel. This acts as an exact projection onto the first $L$ eigenmodes, capturing global geometry while rigorously filtering out fine-grained spatial noise.

Certain weight functions that lead to kernels with known closed form include the chordal distance kernel of the Giné $S_n$ test, the Poisson kernel, and (in terms of sums of special functions) the thin-plate spline kernel of order one Beatson et al. (2018); Jupp (2008).

Having established these rigorous kernel families, the final step is to integrate them into our general discrepancy measures. Because these specific filters yield well-behaved scalar functions of the pairwise cosine similarity $\varphi(c)$, they allow us to transform the abstract statistical tests introduced in Section 4 into exact, computationally tractable training objectives, as we derive in the next section.

## 6 Explicit Forms of the Statistical Tests

Using the rotationally invariant zonal kernels $\varphi(c)$ evaluated on the pairwise cosine similarity $c = x^\top y$, the general discrepancies introduced in Section 4 evaluate analytically. This yields the following deterministic, closed-form objectives (complete derivations are deferred to Appendices B, C and D):

$$D_{\mathrm{MMD}} = \frac{1}{C_{\mathrm{norm/MMD}}} \left( \mathbb{E}_{x,y \sim \hat{p}}[\varphi(c)] - C_{\mathrm{bias/MMD}} \right), \tag{11}$$

$$D_{\mathrm{KSD}} = \frac{1}{C_{\mathrm{norm/KSD}}} \mathbb{E}_{x,y \sim \hat{p}} \left[ \frac{1}{2} \left( (c^2 - 1)\varphi''(c) + c(d-1)\varphi'(c) \right) \right], \tag{12}$$

$$D_{\mathrm{KL}} = \frac{1}{C_{\mathrm{norm/KL}}} \left( \mathbb{E}_{x \sim \hat{p}} \left[ \log \mathbb{E}_{y \sim \hat{p}_{-x}}[\varphi(c)] \right] - C_{\mathrm{bias/KL}} \right). \tag{13}$$

For MMD and KSD, expectations over the empirical distribution $\hat{p}$ are computed using standard V-estimators. For the KL divergence, $\hat{p}_{-x}$ denotes the leave-one-out empirical estimator to prevent trivial self-similarity singularities.

To ensure strict comparability across objectives, each discrepancy is offset and scaled by analytic constants such that a total representation collapse to a Dirac delta distribution (i.e., $c = 1$ for all pairs) yields a worst-case loss of exactly 1. Assuming a normalized base kernel $\varphi(1) = 1$, these constants are defined as follows:

- **MMD:** $C_{\text{bias/MMD}} = \int_{-1}^{1} \varphi(v)\rho_d(v)dv$ is evaluated numerically via highly efficient Gauss-Jacobi quadrature utilizing the hyperspherical marginal density $\rho_d(v) \propto (1 - v^2)^{\frac{d-3}{2}}$, and $C_{\text{norm/MMD}} = 1 - C_{\text{bias/MMD}}$.

- **KSD:** $C_{\text{norm/KSD}} = \frac{d-1}{2}\varphi'(1)$ (with no bias term required due to the zero-score property of the uniform target).

- **KL:** $C_{\text{bias/KL}} = \log |\mathbb{S}^{d-1}|$ and $C_{\text{norm/KL}} = -C_{\text{bias/KL}}$.

In summary, Equations (11)–(13) provide the exact, closed-form objectives that replace stochastic 1D approximations. Computationally, for a batch size $B$, our exact formulation scales as $\mathcal{O}(B^2)$, whereas sliced methods scale as $\mathcal{O}(B|\mathcal{A}|)$, where $|\mathcal{A}|$ is the number of random projections. As a result, exact tests are strictly cheaper to compute when the batch size satisfies $B < |\mathcal{A}|$. Because standard sliced implementations default to $|\mathcal{A}| = 1024$ projections (Nicollier et al., 2026), our exact objectives require less compute time for reasonable batch sizes, but become more expensive when scaling $B$ beyond this threshold.

## 7 Experimental Evaluation

We empirically validate our exact, full-dimensional statistical tests across a diverse suite of benchmarks. Through these experiments, we aim to answer three practical questions: (1) Does analytically bypassing projection variance (via the induced kernel) consistently improve representation quality over stochastic baselines like SUSReg? (2) How do the different spectral kernels (Induced SUSReg, Heat, and Bandlimited) compare empirically? (3) How do the different exact tests (MMD, KSD, and KL-based) behave depending on the underlying topology of the dataset (e.g., clustered object classes versus unclustered continuous textures)?

Unless otherwise specified, all models employ a projection head outputting to $\mathbb{S}^{255}$. To ensure fair comparisons, we maintain consistent hyperparameter configurations across methods. Specifically, we set the regularization weight to $\lambda = 0.05$ for MMD and KSD, strictly matching the established baseline in SPHERE-JEPA (Nicollier et al., 2026). For the KL divergence, we set $\lambda = 0.5$; this value is motivated by its structural similarity to the InfoNCE objective, which, when explicitly decomposed into an invariance loss and a repulsive contrastive term, naturally induces an effective regularization weight of 0.5. Furthermore, for all experiments utilizing the heat kernel, we set the temperature to $t = 5/d$ for MMD and KSD, and $t = 2/d$ for the KDE-based KL divergence; an ablation justifying these specific thermal operating points is provided in Appendix E.

### 7.1 Standard Pretraining (ImageNet-100 & Galaxy10)

We first evaluate the quality of the representations by pretraining a ResNet-18 (He et al., 2016) on ImageNet-100 (Deng et al., 2009) for 200 epochs and a ResNet-50 (He et al., 2016) on the Galaxy10 dataset (Leung and Bovy, 2018) for 200 epochs (see Figure 4 for dataset samples). Because our exact regularizers estimate the hyperspherical distribution directly from the empirical batch, maintaining consistent statistics is critical; we therefore strictly fix the global batch size to 256 across all runs. We subsequently assess the semantic quality of the frozen features using standard linear probing and $k$-NN classification.

As shown in Table 1, the exact deterministic regularizers consistently outperform the stochastic SUSReg baseline across both datasets. On ImageNet-100, MMD and KSD variants improve linear probing accuracy by up to 1.3% and $k$-NN accuracy by up to 1.6%. On Galaxy10, the performance gains reach up to 5.1% in linear probing and 4.1% in $k$-NN classification.

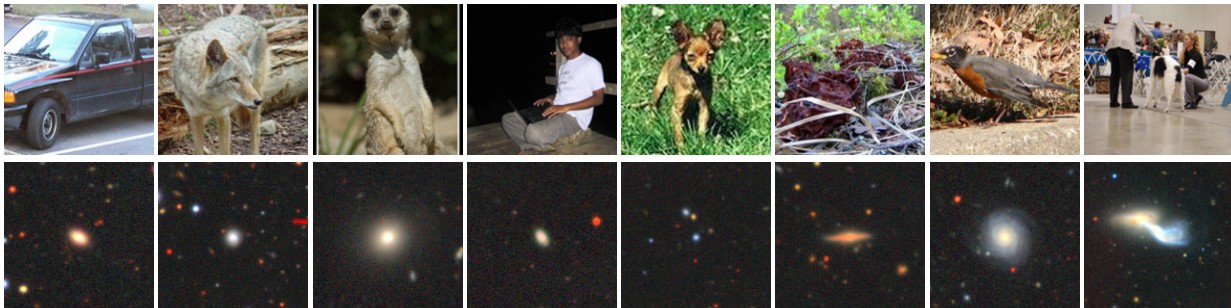

Figure 4: Examples of images from the datasets used in our experiments. The first row shows samples from ImageNet100, and the second row shows samples from Galaxy10.

Table 1: Linear probing and $k$-NN classification accuracy (%) for standard pretraining on ImageNet-100 and Galaxy10. Values are reported as mean ± sample standard deviation over seeds. Bold values indicate the best mean performance within each dataset and protocol.

| Method | ImageNet-100 | | Galaxy10 | |
|---|---|---|---|---|
| | Linear | $k$-NN | Linear | $k$-NN |
| SUSReg (Stochastic Baseline) | $71.22 \pm 0.58$ | $65.67 \pm 0.56$ | $72.13 \pm 1.39$ | $68.67 \pm 1.04$ |
| MMD (Induced SUSReg $\bar{k}$) | $72.17 \pm 0.40$ | $67.02 \pm 0.23$ | $76.78 \pm 0.31$ | $71.67 \pm 0.89$ |
| MMD (Heat, $t = 5/d$) | $72.19 \pm 0.26$ | $67.21 \pm 0.22$ | $76.31 \pm 0.56$ | $71.91 \pm 0.62$ |
| MMD (Bandlimited, $L = 2$) | $72.26 \pm 0.44$ | $\mathbf{67.25 \pm 0.43}$ | $76.21 \pm 0.49$ | $\mathbf{72.73 \pm 1.00}$ |
| KSD (Heat, $t = 5/d$) | $\mathbf{72.55 \pm 0.29}$ | $66.71 \pm 0.31$ | $76.76 \pm 0.40$ | $72.03 \pm 0.57$ |
| KSD (Bandlimited, $L = 2$) | $72.19 \pm 0.38$ | $66.91 \pm 0.14$ | $\mathbf{77.21 \pm 0.55}$ | $72.70 \pm 0.77$ |
| KL (Heat) | $67.72 \pm 0.21$ | $62.09 \pm 0.42$ | $75.76 \pm 0.99$ | $70.29 \pm 0.85$ |

The direct comparison between SUSReg and MMD equipped with the analytically induced kernel $\bar{k}$ isolates the impact of projection noise, as both objectives share the same continuous limit. Replacing the stochastic 1D projections with the exact evaluation yields a performance increase of 1.0% on ImageNet-100 and 4.7% on Galaxy10 for linear probing. Among the spectral kernels, the Bandlimited filter ($L = 2$) slightly outperforms the smooth Heat kernel and the Induced kernel on these multi-class tasks. Conversely, the KDE-based KL divergence underperforms on both datasets, scoring below the baseline on ImageNet-100. This indicates that the continuous repulsion enforced by the KL objective penalizes the local macroscopic clustering necessary for category-level semantic classification.

### 7.2 Instance-Level Texture Retrieval

Given that the KL divergence underperformed on class-heavy datasets like ImageNet, we hypothesize that its continuous surrogate formulation might be optimally suited for purely unclustered distributions. To verify this, we investigate instance-level discrimination via nearest-neighbor retrieval on four procedural texture datasets (Cloud, Disk, Flake, Wood) introduced in SPHERE-JEPA (Nicollier et al., 2026) (see Figure 5 for dataset samples). Following their exact evaluation protocol, this benchmark provides a strictly continuous visual domain where discrete object classes do not exist. Detailed per-dataset results are provided in Appendix F.

Remarkably, as detailed in Table 2, the KDE-based KL divergence strongly dominates its MMD and KSD counterparts in this scenario. KL (Heat) achieves a top-1 retrieval accuracy (Recall@1) of 95.3%, representing a substantial +6.6 point improvement over the stochastic SUSReg baseline (88.7%) and outperforming the best alternative continuous metric, KSD (Heat), by +3.3 points. Similar margins are observed across the

Table 2: Average procedural texture retrieval performance across four datasets. The continuous KL divergence strongly dominates on these unclustered domains.

| Method | Recall@1 | Recall@3 | Recall@5 | mAP | mAP (emb) |
|---|---|---|---|---|---|
| SUSReg (Stochastic Baseline) | 88.7 | 96.1 | 97.7 | 92.6 | 91.4 |
| MMD (Induced SUSReg $\bar{k}$) | 89.1 | 96.1 | 97.8 | 92.9 | 91.1 |
| MMD (Heat, $t = 5/d$) | 91.4 | 97.0 | 98.3 | 94.4 | 93.2 |
| MMD (Bandlimited, $L = 2$) | 91.5 | 96.6 | 97.7 | 94.3 | 93.7 |
| KSD (Heat, $t = 5/d$) | 92.0 | 97.1 | 98.2 | 94.8 | 93.5 |
| KSD (Bandlimited, $L = 2$) | 91.6 | 96.6 | 97.7 | 94.3 | 93.6 |
| KL (Heat) | **95.3** | **97.6** | **98.3** | **96.7** | **96.3** |

mean Average Precision (mAP) metrics, demonstrating robust improvements in both the projection head and the underlying embeddings.

This performance gap directly validates our geometric intuition. Because procedural textures lack semantic element clusters, the continuous KL divergence optimally regularizes the manifold by uniformly repelling all instances individually, leading to exact hyperspherical uniformity. Conversely, point-based integral metrics like MMD and KSD inherently accommodate and even encourage macroscopic local clustering. While this behavior is highly beneficial for multi-class discrimination (e.g., ImageNet) where semantic grouping is desirable, it proves fundamentally suboptimal for continuous, instance-level texture spaces.

## 8 Conclusion

In this work, we first demonstrated that sliced statistical regularizers for hyperspherical uniformity are strictly suboptimal compared to their analytically integrated Maximum Mean Discrepancy (MMD) equivalent. Building on this insight, we introduced a broader family of exact, full-dimensional regularizers based on MMD, Kernel Stein Discrepancy (KSD), and the Kullback-Leibler (KL) divergence. To avoid spatial bias, we grounded these tests in spectral theory, explicitly motivating the use of two canonical kernels: the Heat and Bandlimited filters.

Empirically, our exact formulations confirmed that completely bypassing Monte Carlo projection noise systematically improves representation quality across all evaluated datasets. On object-centric domains, MMD and KSD equipped with our canonical spectral kernels achieved similar state-of-the-art performance, outperforming both the standard stochastic baseline and its exact SUSReg-induced kernel. Finally, we revealed how the choice of statistical discrepancy fundamentally shapes the learned topology: while MMD and KSD inherently accommodate the local clustering necessary for distinct classes, the continuous KDE-based KL divergence optimally forces fine-grained instance repulsion, yielding vastly superior performance on unclustered procedural textures.

Despite these theoretical and empirical advantages, it is important to contextualize the computational trade-offs of exact manifold regularizers. Because our full-dimensional tests rely on exact pairwise kernel evaluations, their computational complexity scales quadratically with the global batch size, $\mathcal{O}(B^2)$. In contrast, stochastic sliced methods like SUSReg compute 1D statistics and scale linearly with respect to the batch size, $\mathcal{O}(B|\mathcal{A}|)$, where $|\mathcal{A}|$ is the number of random projections. Consequently, while our exact regularizers strictly dominate in standard training regimes, projection-based approaches remain a highly practical and necessary alternative when scaling SSL architectures to massive batch sizes where quadratic complexity becomes computationally prohibitive.

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

## A Formal Equivalence and Analytical Integration of Sliced Discrepancies

In this section, we provide the formal proofs supporting the theoretical claims made in Section 3. Specifically, we establish that the expected value of the sliced Epps-Pulley (EP) test—which serves as the core statistical regularizer in stochastic methods like SUSReg—is equivalent to a full-dimensional Maximum Mean Discrepancy (MMD) natively operating on $\mathbb{S}^{d-1}$.

We proceed in three logical steps: (1) we show via Bochner's theorem that the 1D EP test is equivalent to a 1D MMD; (2) we prove via Fubini's theorem that integrating this 1D MMD over all uniform random projections yields a valid full-dimensional MMD; and (3) we geometrically derive the explicit 1D integral form of the induced kernel $\bar{k}$ introduced in the main text.

**Lemma 1** (The 1D Epps-Pulley Test is a 1D MMD [p. 4 (Rustamov, 2021)]). *Let $U$ and $V$ be scalar random variables with characteristic functions $\varphi_U$ and $\varphi_V$. The Epps-Pulley discrepancy is defined as*

$$\mathrm{EP}(U,V) = \int_{\mathbb{R}} \left|\varphi_U(t) - \varphi_V(t)\right|^2 w(t)\, dt, \tag{14}$$

where $w(t) = \frac{1}{\sqrt{2\pi}} \exp\left(-\frac{t^2}{2}\right)$. *This formulation is equivalent to a Maximum Mean Discrepancy:*

$$\text{EP}(U, V) = \text{MMD}_{k_1}^2(U, V), \tag{15}$$

*where $k_1(u, v) = \exp\left(-\frac{(u-v)^2}{2}\right)$ is the standard Gaussian base kernel.*

*Proof.* By Bochner's theorem (Rahimi and Recht, 2007), a continuous, translation-invariant positive definite kernel $k_1(u, v) = \psi(u - v)$ admits a spectral representation via the inverse Fourier transform of a density $w(t)$:

$$\psi(\tau) = \int_{\mathbb{R}} e^{it\tau} \, w(t) \, dt.$$

Specifically, the standard Gaussian density $w(t) = \frac{1}{\sqrt{2\pi}} \exp(-t^2/2)$ is exactly the inverse Fourier transform of the standard Gaussian base kernel $k_1(u, v) = \exp\left(-\frac{(u-v)^2}{2}\right)$. For such translation-invariant kernels, the squared MMD between the distributions of $U$ and $V$ can be evaluated directly in the frequency domain as the weighted $L^2$ distance between their characteristic functions:

$$\text{MMD}_{k_1}^2(U, V) = \int_{\mathbb{R}} \left| \varphi_U(t) - \varphi_V(t) \right|^2 w(t) \, dt.$$

This frequency-domain formulation natively matches the exact definition of the EP test, concluding the proof. □

**Lemma 2** (Expected Sliced MMD is a Full-Dimensional MMD [p. 4, Eq. (4)-(6)(Kolouri et al., 2022)). *Let $X$ and $Y$ be random variables in $\mathbb{R}^d$, and let $k_1$ be a continuous positive definite kernel on $\mathbb{R}$. Define the integrated kernel on $\mathbb{R}^d \times \mathbb{R}^d$ as:*

$$\bar{k}(x, y) = \mathbb{E}_{a \sim \text{Unif}(\mathbb{S}^{d-1})} \left[ k_1(a^\top x, a^\top y) \right].$$

*Then $\bar{k}$ is a valid positive definite kernel on $\mathbb{R}^d$, and the expected 1D MMD over random uniform projections is identically equal to the MMD induced by $\bar{k}$:*

$$\mathbb{E}_{a \sim \text{Unif}(\mathbb{S}^{d-1})} \left[ \text{MMD}_{k_1}^2(a^\top X, a^\top Y) \right] = \text{MMD}_{\bar{k}}^2(X, Y).$$

**Remark 1.** *It is worth noting that the integrated kernel $\bar{k}$ on $\mathbb{R}^d$ inherits not just the positive-definiteness of the base kernel $k_1$, but also its characteristic property. It is an easy matter to verify that $\bar{k}$ is characteristic when $k_1$ is directly, using the Cramér-Wold theorem; it follows also as a special case of [p. 4, Proposition 1 (Nadjahi et al., 2020)]. Thus, the induced full-dimensional MMD divergence remains a (definite) metric on probability measures if $k_1$ is.*

*Proof.* Since $k_1$ is positive definite on $\mathbb{R}$, for any fixed direction $a \in \mathbb{S}^{d-1}$, the function $k_a(x, y) := k_1(a^\top x, a^\top y)$ is a positive definite kernel on $\mathbb{R}^d$. Indeed, for any $x_1, \ldots, x_n \in \mathbb{R}^d$ and coefficients $c_1, \ldots, c_n \in \mathbb{R}$,

$$\sum_{i,j} c_i c_j k_a(x_i, x_j) = \sum_{i,j} c_i c_j k_1(a^\top x_i, a^\top x_j) \geq 0.$$

Because the expectation of positive definite kernels remains positive definite, the integrated kernel $\bar{k}(x, y) = \mathbb{E}_a[k_a(x, y)]$ is a valid positive definite kernel on $\mathbb{R}^d$.

To prove the equality of the discrepancies, we expand the expected one-dimensional MMD:

$$\mathbb{E}_a \left[ \text{MMD}_{k_1}^2(a^\top X, a^\top Y) \right] = \mathbb{E}_a \Big[ \mathbb{E}_{X,X'}[k_1(a^\top X, a^\top X')] + \mathbb{E}_{Y,Y'}[k_1(a^\top Y, a^\top Y')]$$
$$- 2 \, \mathbb{E}_{X,Y}[k_1(a^\top X, a^\top Y)] \Big].$$

By Fubini's theorem, we can safely exchange the expectations over the random directions $a$ and the random variables $X, Y$:

$$\mathbb{E}_a\big[\mathrm{MMD}^2_{k_1}(a^\top X, a^\top Y)\big] = \mathbb{E}_{X,X'}\big[\mathbb{E}_a[k_1(a^\top X, a^\top X')]\big] + \mathbb{E}_{Y,Y'}\big[\mathbb{E}_a[k_1(a^\top Y, a^\top Y')]\big]$$
$$- 2\,\mathbb{E}_{X,Y}\big[\mathbb{E}_a[k_1(a^\top X, a^\top Y)]\big].$$

Substituting the definition of the integrated kernel $\bar{k}$ into the expectations yields exactly the full-dimensional formulation $\mathrm{MMD}^2_{\bar{k}}(X, Y)$. $\qquad\square$

**Derivation of the Explicit Integral Form.** By sequentially applying Lemma 1 and Lemma 2, we have established that the expected sliced objective resolves exactly to $\mathbb{E}_a[\mathrm{EP}(a^\top X, a^\top Y)] = \mathrm{MMD}^2_{\bar{k}}(X, Y)$. To compute this deterministically in practice, we must derive the explicit form of the induced kernel $\bar{k}(x, y)$.

For unit vectors $x, y \in \mathbb{S}^{d-1}$, the standard Gaussian base kernel evaluates to:

$$k_1(a^\top x, a^\top y) = \exp\left(-\frac{(a^\top(x-y))^2}{2}\right).$$

Let $c = x^\top y$ denote the cosine similarity. On the unit sphere, the distance between embeddings is $\|x - y\| = \sqrt{2(1-c)}$. We can therefore factorize the random projection along the unit direction $u = \frac{x-y}{\|x-y\|}$ as:

$$a^\top(x-y) = \|x - y\|(a^\top u) = \sqrt{2(1-c)} \cdot t,$$

where $t = a^\top u$. Since the vector $a$ is drawn uniformly from $\mathbb{S}^{d-1}$ and $u$ is fixed, the scalar projection $t \in [-1, 1]$ natively follows the hyperspherical marginal distribution, whose probability density is:

$$\rho_d(t) = \frac{\Gamma(d/2)}{\sqrt{\pi}\Gamma((d-1)/2)}(1-t^2)^{\frac{d-3}{2}}.$$

Substituting this geometric projection back into the kernel expectation, the multidimensional integration over the random directions $a$ simplifies into a single 1D integral. The exponent simplifies cleanly since $\frac{1}{2}(\sqrt{2(1-c)}t)^2 = (1-c)t^2$, yielding:

$$\bar{k}(x, y) = \int_{-1}^{1} \exp\left(-(1-c)t^2\right)\rho_d(t)\, dt.$$

This derivation confirms that the induced kernel is strictly rotationally invariant—depending solely on the cosine similarity $c$—and matches the explicit equation presented in Section 3. Furthermore, this form naturally justifies the use of Gauss-Jacobi quadrature for fast, deterministic, and fully differentiable evaluations.

## B  Kernel Stein Discrepancy on the Hypersphere

In this appendix, we derive a closed-form expression of the Kernel Stein Discrepancy (KSD) for the uniform distribution on the hypersphere $\mathbb{S}^{d-1}$, yielding the explicit objective $D_{\mathrm{KSD}}$ presented in Section 6.

While the theoretical properties and consistency of this generalized Riemannian KSD have been established in Qu and Vemuri (2025), we provide here a self-contained derivation explicitly tailored to our uniformity test.

**The Riemann-Stein Framework.** Building upon the geometric Stein kernel framework (Barp et al., 2022) and its directional formulation (Xu and Matsuda, 2020), let $\mathcal{M}$ be a Riemannian manifold and $q$ a target density. The main idea behind the Stein method is to find an operator $\mathcal{A}$ such that $\mathbb{E}_{x \sim q}[\mathcal{A}f(x)] = 0$ for any suitable test function $f$. By Stokes theorem, on a compact manifold without boundary, the integral of the divergence of any vector field is zero. To construct our operator $\mathcal{A}_X$, we consider the density-weighted

vector field $Y = f \cdot q \cdot X$, where $f$ is a scalar test function and $X$ is a fixed vector field. Expanding its divergence via the standard product rule yields:

$$\text{div}(fqX) = \langle \nabla(fq), X \rangle + fq \, \text{div}(X)$$
$$= q \langle \nabla f, X \rangle + f \langle \nabla q, X \rangle + fq \, \text{div}(X).$$

Factoring out $q$ and using the identity $\nabla q = q \nabla \log q$, we obtain $\text{div}(fqX) = q \mathcal{A}_X f$, which defines the general Stein operator acting on $f$ along $X$:

$$\mathcal{A}_X f(x) = \langle X(x), \nabla f(x) \rangle + \big( \text{div}(X)(x) + \langle X(x), \nabla \log q(x) \rangle \big) f(x). \tag{16}$$

Since $\int_{\mathcal{M}} \text{div}(fqX) d\mu = 0$, it immediately follows that the expectation under $q$ is zero: $\mathbb{E}_q[\mathcal{A}_X f] = 0$.

**Strategic Choice of Vector Field.**  The general Stein operator defined in (16) is valid for any smooth vector field $X$. However, evaluating this operator depends on computing the divergence $\text{div}(X)$, which often leads to numerically unstable expressions (such as Jacobian derivatives in local spherical coordinates).

To circumvent this issue, we will choose $X$ to be a divergence-free vector field, more specifically a Killing vector field or infinitessimal isometry.

For the hypersphere $\mathbb{S}^{d-1}$, these isometries are generated by the Lie algebra $\mathfrak{so}(d)$ consisting of all skew-symmetric matrices of size $d \times d$. We adopt its canonical orthonormal basis, given by the matrices $E_{ij}$ for $1 \leq i < j \leq d$:

$$E_{ij} = \frac{1}{\sqrt{2}} \left( e_i e_j^\top - e_j e_i^\top \right) \in \mathfrak{so}(d),$$

/here $e_i$ is the $i$-th canonical basis vector of $\mathbb{R}^d$. Applying these matrices to a point $x \in \mathbb{S}^{d-1}$ yields a basis of Killing vector fields. By setting our generic field to $X(x) = E_{ij}x$, the divergence term in (16) vanishes.

Furthermore, since our target distribution $q$ is the uniform density on $\mathbb{S}^{d-1}$, its log-derivative vanishes ($\nabla \log q \equiv 0$). Equation (16) therefore reduces to:

$$\mathcal{A}_{ij} f(x) = (E_{ij}x)^\top \nabla_x f(x). \tag{17}$$

**Constructing the Stein Kernel.**  The mechanism of the Kernel Stein Discrepancy established by Chwialkowski et al. (2016), evaluates the discrepancy by taking the supremum over a unit ball in a Reproducing Kernel Hilbert Space (RKHS). Thanks to the reproducing property, this supremum evaluates in closed form to the expectation of a symmetric Stein kernel $k_q(x, y)$.

This Stein kernel is constructed by applying the Stein operator to a base scalar reproducing kernel $k(x, y)$ on both variables. Because this directional approach utilizes a set of independent operators—one for each skew-symmetric matrix in the Lie algebra $\mathfrak{so}(d)$—the overall Stein kernel on the hypersphere is formed by applying the double operator along each valid direction and summing the results over the entire basis:

$$k_q(x, y) = \sum_{1 \leq i < j \leq d} \left( \mathcal{A}_{ij}^x \mathcal{A}_{ij}^y k \right)(x, y). \tag{18}$$

Let us expand the inner term. First, applying the operator with respect to $y$ yields a row vector:

$$\mathcal{A}_{ij}^y k(x, y) = (E_{ij}y)^\top \nabla_y k(x, y) = \nabla_y^\top k(x, y)(E_{ij}y).$$

Next, applying the operator with respect to $x$ to this result requires the Hessian of the kernel, obtained via standard multivariable calculus:

$$\left( \mathcal{A}_{ij}^x \mathcal{A}_{ij}^y k \right)(x, y) = (E_{ij}x)^\top \left[ \nabla_x \big( \nabla_y^\top k(x, y) \big) \right](E_{ij}y). \tag{19}$$

**Specialization to Radial Kernels.** We now specialize to radial (zonal) kernels on the sphere, which only depend on the inner product $c := x^\top y$. Let $k(x, y) = \phi(c)$ for $x, y \in \mathbb{S}^{d-1}$, where $\phi : [-1, 1] \to \mathbb{R}$ is twice differentiable.

Using this notation, we have $\phi'(c) = \nabla_y k(x, y)x$.

Differentiating with respect to $x$ yields the Hessian:

$$\nabla_x \nabla_y^\top k(x, y) = \phi''(c)yx^\top + \phi'(c)I. \tag{20}$$

**Evaluating the Sum over the Lie Algebra.** Substituting (20) back into (19), the term for a single basis vector becomes:

$$
\begin{aligned}
\left(\mathcal{A}_{ij}^x \mathcal{A}_{ij}^y k\right)(x, y) &= (E_{ij}x)^\top \left(\phi''(c)yx^\top + \phi'(c)I\right)(E_{ij}y) \\
&= \phi''(c)\left((E_{ij}x)^\top y\right)\left(x^\top E_{ij}y\right) + \phi'(c)\left((E_{ij}x)^\top (E_{ij}y)\right).
\end{aligned}
\tag{21}
$$

To compute the sum over $1 \le i < j \le d$, an expansion using linear algebra yields the matrix $M$:

$$M(x, y) := \sum_{1 \le i < j \le d} (E_{ij}x)(E_{ij}y)^\top = \frac{1}{2}\left(c\,I - yx^\top\right). \tag{22}$$

We can now evaluate the two terms in (21) using $M$:

**The $\phi''(c)$ term (Quadratic Form):** Since $E_{ij}$ is skew-symmetric ($E_{ij}^\top = -E_{ij}$), we can manipulate the transposes: $(E_{ij}x)^\top y = -x^\top E_{ij}y = y^\top E_{ij}x$. Moreover, the scalar $x^\top E_{ij}y$ equals its transpose $-y^\top E_{ij}x$. The product thus becomes:

$$\left((E_{ij}x)^\top y\right)\left(x^\top E_{ij}y\right) = (y^\top E_{ij}x)(-y^\top E_{ij}x) = -(y^\top E_{ij}x)^2.$$

Summing this quadratic expression is equivalent to evaluating the form $y^\top Mx$ on the matrix $M$ defined in (22), because $\sum_{i<j}(y^\top E_{ij}x)(y^\top E_{ij}^\top x) = y^\top Mx$. Evaluating this explicitly yields:

$$y^\top Mx = y^\top \left(\frac{1}{2}(c\,I - yx^\top)\right)x = \frac{1}{2}\left(c(y^\top x) - (y^\top y)(x^\top x)\right) = \frac{1}{2}(c^2 - 1). \tag{23}$$

**The $\phi'(c)$ term (Trace):** The dot product of two vectors $u^\top v$ is equal to the trace of their outer product $\mathrm{Tr}(vu^\top)$. Therefore:

$$\sum_{1 \le i < j \le d} (E_{ij}x)^\top (E_{ij}y) = \sum_{1 \le i < j \le d} \mathrm{Tr}\left((E_{ij}y)(E_{ij}x)^\top\right) = \mathrm{Tr}(M).$$

Taking the trace of $M$ from (22), $\mathrm{Tr}(I) = d$ and $\mathrm{Tr}(yx^\top) = x^\top y = c$, yields:

$$\mathrm{Tr}(M) = \mathrm{Tr}\left(\frac{1}{2}(c\,I - yx^\top)\right) = \frac{1}{2}(cd - c) = \frac{c(d-1)}{2}. \tag{24}$$

**Closed Form.** Combining these two evaluated sums, we obtain our remarkably simple closed form for the Stein kernel of the uniform distribution on $\mathbb{S}^{d-1}$, depending on the first two derivatives of the radial kernel $\phi$:

$$k_q(x, y) = \frac{1}{2}\left[(c^2 - 1)\phi''(c) + c(d-1)\phi'(c)\right], \qquad c = x^\top y. \tag{25}$$

**Normalization for Representation Learning.** Substituting this closed-form Stein kernel back into the expectation over the empirical distribution $\hat{p}$ yields our raw KSD objective. As established in Section 6, we scale this discrepancy such that a worst-case representation collapse evaluates exactly to 1. In the event of

collapse ($x = y$, yielding $c = 1$), the term $(c^2 - 1)\phi''(c)$ vanishes, and the expectation evaluates to exactly $\frac{d-1}{2}\phi'(1)$.

By defining the normalization constant $C_{\text{norm/KSD}} = \frac{d-1}{2}\phi'(1)$, we scale the objective such that a complete point collapse ($c = 1$) evaluates to exactly 1, yielding the final explicit form $D_{\text{KSD}}$ presented in Section 6:

$$D_{\text{KSD}} = \frac{1}{C_{\text{norm/KSD}}}\mathbb{E}_{x,y\sim\hat{p}}\left[\frac{1}{2}\Big((c^2 - 1)\phi''(c) + c(d-1)\phi'(c)\Big)\right]. \tag{26}$$

## C   Maximum Mean Discrepancy on the Hypersphere

In this appendix, we detail the derivation of the Maximum Mean Discrepancy (MMD) (Gretton et al., 2012) on $\mathbb{S}^{d-1}$ and specialize it to the rotationally invariant zonal kernels introduced in Section 5, yielding the explicit objective presented in Section 6.

**Maximum Mean Discrepancy on a Manifold.**   Let $\mu$ be the Haar measure on $\mathbb{S}^{d-1}$, and let

$$k : \mathbb{S}^{d-1} \times \mathbb{S}^{d-1} \to \mathbb{R}$$

be a positive definite kernel with an associated reproducing kernel Hilbert space (RKHS) $\mathcal{H}$. For two probability densities $p$ and $q$ on $\mathbb{S}^{d-1}$ (with respect to $\mu$), the squared Maximum Mean Discrepancy is defined as the distance between their mean embeddings:

$$\text{MMD}^2(p, q) = \|\mu_p - \mu_q\|_{\mathcal{H}}^2,$$

where the mean embedding of a distribution $r$ is given by $\mu_r = \int_{\mathbb{S}^{d-1}} k(\cdot, x)\, r(x)\, \mu(dx)$.

Expanding the RKHS norm yields the classical expression:

$$\text{MMD}^2(p, q) = \mathbb{E}_{x,x'\sim p}[k(x, x')] + \mathbb{E}_{y,y'\sim q}[k(y, y')] - 2\,\mathbb{E}_{x\sim p, y\sim q}[k(x, y)]. \tag{27}$$

**Zonal Kernels and the Uniform Target.**   Following Section 4, we evaluate the discrepancy between the empirical distribution of the embeddings, $p = \hat{p}$, and the uniform target distribution, $q = \text{Unif}(\mathbb{S}^{d-1})$. Furthermore, as established in Section 5, we restrict our focus to rotationally invariant zonal kernels of the form:

$$k(x, y) = \varphi(c), \qquad \text{where} \quad c := x^\top y \in [-1, 1].$$

Substituting these into Equation (27), the empirical MMD becomes:

$$\text{MMD}^2(\hat{p}, q) = \mathbb{E}_{x,x'\sim\hat{p}}\big[\varphi(x^\top x')\big] + \mathbb{E}_{y,y'\sim q}\big[\varphi(y^\top y')\big] - 2\,\mathbb{E}_{x\sim\hat{p}, y\sim q}\big[\varphi(x^\top y)\big]. \tag{28}$$

Because $q$ is uniform, by rotational invariance, if $y \sim q$ and $x \in \mathbb{S}^{d-1}$ is a fixed vector, the inner product $v = x^\top y$ follows a known distribution with density:

$$\rho_d(v) = \frac{(1 - v^2)^{\frac{d-3}{2}}}{B\left(\frac{1}{2}, \frac{d-1}{2}\right)}, \qquad v \in [-1, 1],$$

where $B(\cdot, \cdot)$ denotes the Beta function. Consequently, expectations involving the uniform target reduce to a tractable one-dimensional integral over $[-1, 1]$. We define this analytic bias constant as:

$$C_{\text{bias/MMD}} := \int_{-1}^{1} \varphi(v)\rho_d(v)dv.$$

Because both the target-target expectation and the cross-term expectation integrate over the uniform measure $q$, they simplify identically to this constant:

$$\mathbb{E}_{y,y'\sim q}\big[\varphi(y^\top y')\big] = C_{\text{bias/MMD}},$$

$$\mathbb{E}_{x\sim\hat{p}, y\sim q}\big[\varphi(x^\top y)\big] = C_{\text{bias/MMD}}.$$

Substituting these identities back into Equation (28) yields the unnormalized objective:

$$\text{MMD}^2(\hat{p}, q) = \mathbb{E}_{x,x'\sim\hat{p}}\big[\varphi(x^\top x')\big] - C_{\text{bias/MMD}}. \tag{29}$$

**Normalization for Representation Learning.** To ensure comparable gradients and bounded objectives across different kernels and dimensionalities, we scale this raw discrepancy to obtain a worst-case collapse value of exactly 1. Total representation collapse occurs when all embeddings are mapped to the same point on the hypersphere (i.e., $x = x'$, yielding $x^\top x' = 1$).

Since our kernels are systematically normalized such that $\varphi(1) = 1$, the maximum possible value of the unnormalized MMD is $1 - C_{\text{bias/MMD}}$. By defining the normalization constant $C_{\text{norm/MMD}} = 1 - C_{\text{bias/MMD}}$, we obtain the final explicit regularization objective $D_{\text{MMD}}$ presented in Section 6:

$$D_{\text{MMD}} = \frac{1}{C_{\text{norm/MMD}}} \left( \mathbb{E}_{x,x' \sim \hat{p}}[\varphi(c)] - C_{\text{bias/MMD}} \right). \tag{30}$$

# D Kullback–Leibler Divergence on the Hypersphere

In this appendix, we detail the derivation of the Kullback–Leibler (KL) divergence on the hypersphere $\mathbb{S}^{d-1}$ and show how a Kernel Density Estimation (KDE) approach leads to the explicit, closed-form objective $D_{\text{KL}}$ presented in Section 6.

**Kullback–Leibler Divergence on a Manifold.** Let $\mu$ be the Haar measure on $\mathbb{S}^{d-1}$. For two probability densities $p$ and $q$ on $\mathbb{S}^{d-1}$ (with respect to $\mu$), the Kullback–Leibler divergence is defined as:

$$\text{KL}(p\|q) = \int_{\mathbb{S}^{d-1}} p(x) \log \frac{p(x)}{q(x)} \, \mu(dx). \tag{31}$$

Equivalently, writing the expectation with respect to $p$, we have:

$$\text{KL}(p\|q) = \mathbb{E}_{x \sim p}[\log p(x) - \log q(x)]. \tag{32}$$

**Uniform Reference Distribution.** We now specialize to the case where the target reference distribution is the uniform measure on the hypersphere, $q = \text{Unif}(\mathbb{S}^{d-1})$. The density of the uniform distribution with respect to the Haar measure is constant and equal to $q(x) = 1/|\mathbb{S}^{d-1}|$, where

$$|\mathbb{S}^{d-1}| = \frac{2\pi^{d/2}}{\Gamma(d/2)}$$

denotes the surface area of the hypersphere. Substituting this constant density into Equation (32) yields:

$$\text{KL}(p\|q) = \mathbb{E}_{x \sim p}\left[ \log p(x) \right] + \log |\mathbb{S}^{d-1}|. \tag{33}$$

This establishes that minimizing the KL divergence to the uniform distribution is fundamentally equivalent to maximizing the differential entropy of $p$. The geometric constant governing this shift is defined in our formulation as $C_{\text{bias/KL}} = \log |\mathbb{S}^{d-1}|$.

**Sample-Based Approximations and Gradient Stability.** In practical representation learning, we only have access to a discrete empirical distribution $\hat{p}$ derived from a finite minibatch of embeddings. Because the exact differential entropy $\mathbb{E}_{x \sim \hat{p}}[\log \hat{p}(x)]$ diverges for a discrete sum of Dirac deltas, we must rely on continuous, sample-based approximations.

While some existing methods (such as the KoLeo estimator (Sablayrolles et al., 2019)) approximate this entropy using nearest-neighbor distances, we explicitly discard this approach. The reliance on a hard min operator to identify the closest neighbor inherently induces highly non-smooth, unstable gradients during backpropagation, severely destabilizing optimization.

Instead, to obtain a robust, smoothly differentiable, and full-dimensional statistical test that seamlessly integrates with our rotationally invariant kernels $k(x, x') = \varphi(c)$, we construct a continuous surrogate for

Table 3: ImageNet-100 heat-kernel temperature sensitivity. Linear probing and $k$-NN classification accuracy (%) are reported as mean $\pm$ sample standard deviation over seeds.

| Temperature | MMD (Heat) | | KSD (Heat) | |
|---|---|---|---|---|
| | Linear | $k$-NN | Linear | $k$-NN |
| $4/d$ | $72.25 \pm 0.70$ | $66.48 \pm 0.21$ | $53.80 \pm 35.20$ | $49.93 \pm 32.62$ |
| $5/d$ | $72.55 \pm 0.29$ | $66.71 \pm 0.31$ | $72.55 \pm 0.29$ | $66.71 \pm 0.31$ |
| $6/d$ | $71.73 \pm 0.49$ | $66.64 \pm 0.60$ | $72.44 \pm 0.30$ | $67.27 \pm 0.42$ |

the empirical density using Kernel Density Estimation (KDE). To prevent trivial self-similarity singularities—where the density estimate at $x$ is entirely dominated by the kernel evaluating its own embedding ($x^\top x = 1$)—we strictly evaluate this density using a leave-one-out estimator, denoted $\hat{p}_{-x}$:

$$\tilde{p}(x) = \mathbb{E}_{x' \sim \hat{p}_{-x}}\left[\varphi(x^\top x')\right].$$

Replacing the exact density $p(x)$ in Equation (33) with this leave-one-out surrogate yields our unnormalized KDE-based objective.

**Normalization for Representation Learning.** As established in Section 6, we systematically scale our discrepancies such that a worst-case representation collapse evaluates exactly to 1. In the event of total collapse, all normalized embeddings map to the exact same coordinate ($x = x'$, yielding $c = 1$). Because our zonal kernels are calibrated such that $\varphi(1) = 1$, the leave-one-out density estimate $\tilde{p}(x)$ equals 1, and its logarithm evaluates to exactly 0.

To map this boundary condition consistently across all tests, we apply the scaling constant $C_{\text{norm/KL}}$ alongside the analytic bias $C_{\text{bias/KL}}$ defined above. This yields the final explicit, normalized objective $D_{\text{KL}}$ as implemented in Section 6:

$$D_{\text{KL}} = \frac{1}{C_{\text{norm/KL}}} \left( \mathbb{E}_{x \sim \hat{p}} \left[ \log \mathbb{E}_{x' \sim \hat{p}_{-x}}[\varphi(c)] \right] - C_{\text{bias/KL}} \right). \tag{34}$$

## E   Ablation of the Temperature for the Heat Kernel

The regularizing behavior of the heat kernel inherently depends on its scale parameter $t$, which dictates the exponential decay of the spectral weights $w(\lambda_\ell) = e^{-t\lambda_\ell}$. On the hypersphere $\mathbb{S}^{d-1}$, the natural baseline for this time step scales inversely with the dimension, establishing $1/d$ as a unit.

To determine a good smoothing factor, we focus our ablation on the critical region where the kernel's spatial profile exhibits the most significant structural variations. As visualized in Figure 3, this corresponds to the temperature range of $t \in \{4/d, 5/d, 6/d\}$. It is worth noting that for density-based evaluations—such as the KDE approximation used in the KL divergence—the requirements are different. In such cases, a much smaller temperature (strictly evaluated at $2/d$) is necessary to ensure the density approximation remains theoretically valid.

The empirical sensitivity of the MMD and KSD objectives to the temperature parameter is detailed in Table 3 for ImageNet-100. An observation is the optimization instability of the KSD objective at $t = 4/d$, which can lead to representation collapse in some runs. This collapse explains the large variance across random seeds (yielding a standard deviation of $\pm 35.20\%$ for linear probing) observed on ImageNet-100.

## F   Texture Retrieval Evaluation

Following the experimental framework established by SPHERE-JEPA (Nicollier et al., 2026), we evaluate our method on their nonparametric texture retrieval task. This task is specifically designed to assess the

geometry of learned representations: given a query image, the objective is to retrieve another view of the same texture instance among visually similar candidate samples.

We adopt their exact evaluation protocol, procedural datasets, and data augmentation pipeline, which we briefly summarize below.

### F.1 Datasets

We use the four procedural texture datasets introduced in Nicollier et al. (2026): *Disk*, *Cloud*, *Flake*, and *Wood*. As originally described by the authors, each dataset is generated from a distinct stochastic process (e.g., heavy-tailed noise, Brownian motion, Perlin noise).

As illustrated in Figure 5, samples within a given dataset are generated using different random seeds. This yields images that share a similar statistical structure while remaining visually distinct. Following their standard setup, each texture family consists of 10,000 training, 500 validation, and 10,000 test samples.

### F.2 Data Augmentation and View Generation

To generate multiple views from each texture image ($V_g = 2$ views per image by default), we directly apply the stochastic spatial and photometric augmentation pipeline proposed in Nicollier et al. (2026).

As visualized in Figure 6, each view is generated using a random affine transformation (incorporating rotation, translation, scaling, and shear), followed by photometric augmentations such as brightness/contrast adjustments and random erasing. These transformations introduce significant spatial variability while preserving the underlying texture statistics. The exact same augmentation pipeline is applied at test time to ensure consistency.

### F.3 Quantitative Results

Evaluation is performed within mini-batches of size $B = 100$. For each query, similarity is computed against the other samples in the batch, and retrieval performance is measured based on the ranking of these candidates. We report nearest-neighbor retrieval performance using Recall@K ($K \in \{1, 3, 5\}$) and mean Average Precision (mAP). To ensure the regularizers effectively structure the underlying feature space rather than just the projection space, we evaluate both on the projection head outputs (mAP) and directly on the frozen backbone embeddings (mAP emb).

Table 2 summarizes the average performance across all four texture domains. Tables 4, 5, 6, and 7 provide the detailed breakdown for each individual dataset.

**Superiority of Exact Hyperspherical Uniformity.** As shown in Table 2, **KL (Heat)** consistently and significantly outperforms all other methods on average. The performance gains are particularly striking in the top-1 retrieval metric, where KL (Heat) achieves an average Recall@1 of 95.3%, representing a substantial +6.6 absolute points improvement over the stochastic SUSReg baseline (88.7%). This dominant trend is consistent across all individual datasets, where KL (Heat) even nearly saturates performance on the easier domains (e.g., 99.4% Recall@1 on Wood).

**Continuous vs. Stochastic Regularization.** The aggregated results clearly validate our theoretical analysis regarding representation geometry in nonparametric settings. The continuous, full-dimensional regularizers—specifically MMD and KSD utilizing Heat or Bandlimited kernels—consistently yield better instance-level discrimination than both the stochastic baseline (SUSReg) and the induced MMD approach. For instance, replacing the induced MMD with a continuous Heat kernel MMD improves the average Recall@1 by +2.3 points (from 89.1% to 91.4%). This confirms that continuous structural constraints pave the way for the substantial jump achieved by the exact KL penalty.

**Backbone vs. Projection Head.** Finally, comparing the average mAP evaluated on the projection head (96.7% for KL Heat) to the mAP on the frozen backbone embeddings (96.3% for KL Heat) reveals that the structural improvements transfer successfully to the core representations. While there is a standard, slight drop when evaluating the embeddings directly across all methods, models regularized with continuous

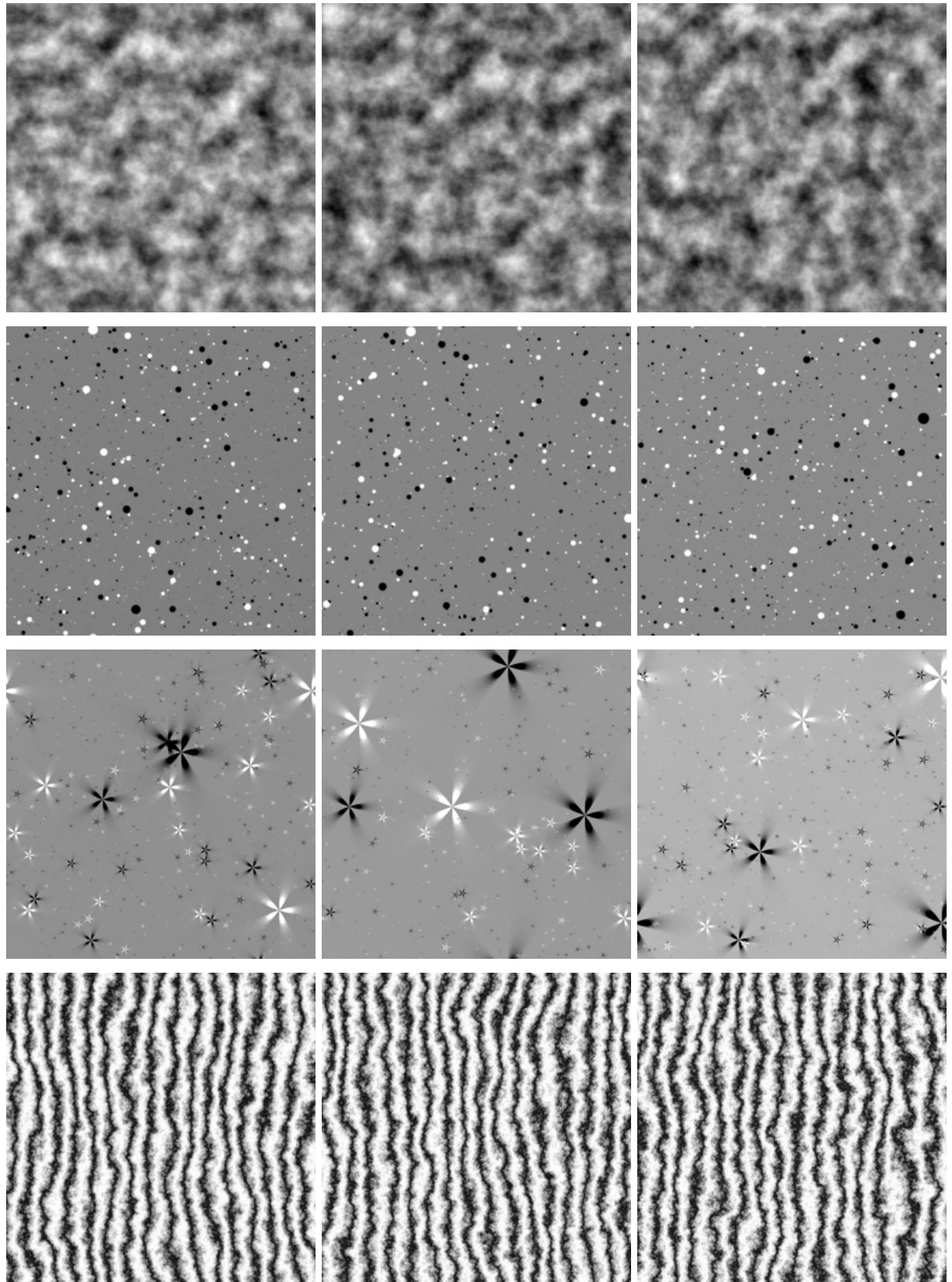

Figure 5: Examples of procedural textures used for retrieval evaluation, as introduced in Nicollier et al. (2026). Each row corresponds to a texture family (top to bottom: cloud, disk, flake, wood). Images within a row are generated from the same stochastic process with different random seeds, resulting in strong statistical similarity but distinct visual realizations.

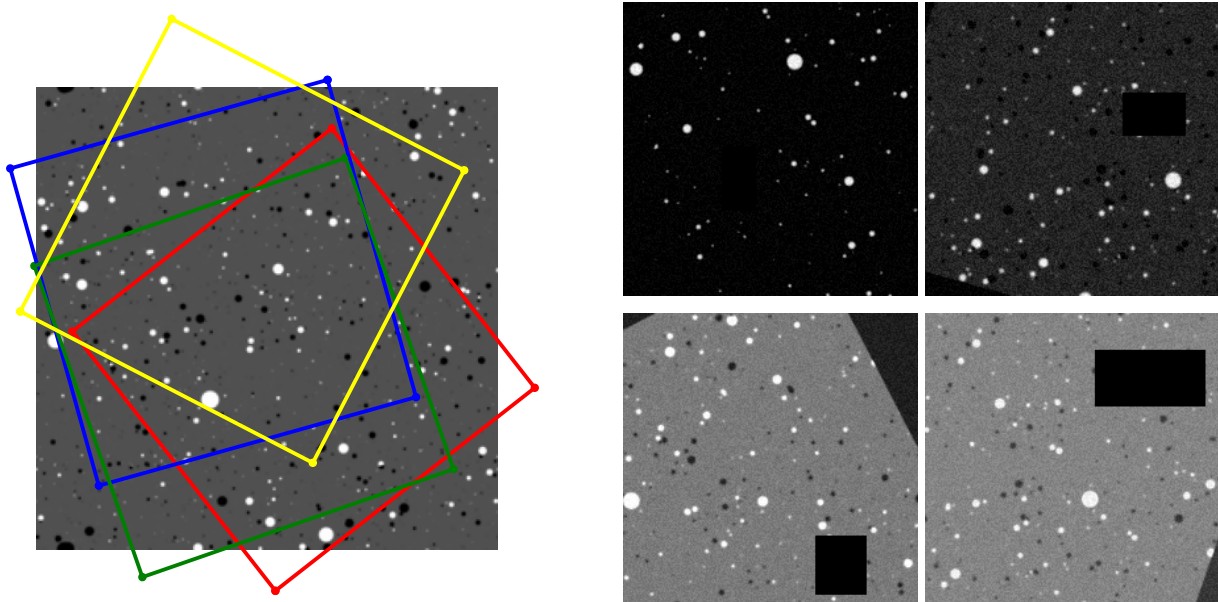

Figure 6: **Left:** effective source regions induced by independently sampled random affine transformations. **Right:** corresponding augmented views obtained after applying the complete affine and photometric transformation pipeline from Nicollier et al. (2026).

metrics maintain excellent embedding performance. This confirms that the uniformly distributed geometry is genuinely learned by the backbone and not merely overfitted within the projection head.

Table 4: Disk Texture

| Method | Recall@1 | Recall@3 | Recall@5 | mAP | mAP (emb) |
|---|---|---|---|---|---|
| SUSReg (Stochastic Baseline) | 88.7 | 95.5 | 97.2 | 92.4 | 90.1 |
| MMD (Induced SUSReg $\bar{k}$) | 87.9 | 95.1 | 97.1 | 91.9 | 88.8 |
| MMD (Heat, $t = 5/d$) | 91.7 | 96.4 | 97.8 | 94.4 | 93.1 |
| MMD (Bandlimited, $L = 2$) | 90.5 | 95.0 | 96.2 | 93.1 | 92.9 |
| KSD (Heat, $t = 5/d$) | 91.9 | 96.5 | 97.7 | 94.5 | 92.5 |
| KSD (Bandlimited, $L = 2$) | 91.1 | 95.7 | 96.7 | 93.7 | 93.3 |
| KL (Heat) | 97.0 | 98.7 | 99.1 | 97.9 | 97.2 |

Table 5: Cloud Texture

| Method | Recall@1 | Recall@3 | Recall@5 | mAP | mAP (emb) |
|---|---|---|---|---|---|
| SUSReg (Stochastic Baseline) | 89.7 | 95.4 | 96.8 | 92.9 | 91.4 |
| MMD (Induced SUSReg $\bar{k}$) | 90.2 | 95.9 | 97.6 | 93.4 | 91.4 |
| MMD (Heat, $t = 5/d$) 93.1 | 97.0 | 98.1 | 95.3 | 94.1 | |
| MMD (Bandlimited, $L = 2$) 93.3 | 97.0 | 97.7 | 95.3 | 95.0 | |
| KSD (Heat, $t = 5/d$) | 93.4 | 97.0 | 97.9 | 95.4 | 94.6 |
| KSD (Bandlimited, $L = 2$) | 91.5 | 95.9 | 96.9 | 94.0 | 93.1 |
| KL (Heat) | 96.3 | 98.0 | 98.5 | 97.3 | 96.2 |

Table 6: Wood Texture

| Method | Recall@1 | Recall@3 | Recall@5 | mAP | mAP (emb) |
|---|---|---|---|---|---|
| SUSReg (Stochastic Baseline) | 93.3 | 99.1 | 99.7 | 96.2 | 95.6 |
| MMD (Induced SUSReg $\bar{k}$) | 95.0 | 99.4 | 99.7 | 97.2 | 96.3 |
| MMD (Heat, $t = 5/d$) | 95.7 | 99.4 | 99.8 | 97.6 | 96.7 |
| MMD (Bandlimited, $L = 2$) | 96.0 | 99.4 | 99.7 | 97.7 | 96.4 |
| KSD (Heat, $t = 5/d$) | 96.9 | 99.6 | 99.9 | 98.3 | 97.3 |
| KSD (Bandlimited, $L = 2$) | 97.4 | 99.6 | 99.8 | 98.5 | 97.7 |
| KL (Heat) | 99.4 | 99.8 | 99.9 | 99.7 | 99.2 |

Table 7: Flake Texture

| Method | Recall@1 | Recall@3 | Recall@5 | mAP | mAP (emb) |
|---|---|---|---|---|---|
| SUSReg (Stochastic Baseline) | 83.0 | 94.3 | 97.3 | 89.1 | 88.6 |
| MMD (Induced SUSReg $\bar{k}$) | 83.5 | 94.2 | 97.3 | 89.3 | 88.0 |
| MMD (Heat, $t = 5/d$) | 85.0 | 95.4 | 97.7 | 90.5 | 89.1 |
| MMD (Bandlimited, $L = 2$) | 86.1 | 95.0 | 97.2 | 90.9 | 90.3 |
| KSD (Heat, $t = 5/d$) | 86.0 | 95.2 | 97.5 | 91.0 | 89.8 |
| KSD (Bandlimited, $L = 2$) | 86.2 | 95.1 | 97.2 | 91.0 | 90.3 |
| KL (Heat) | 88.6 | 94.0 | 95.8 | 91.7 | 92.5 |

