# OpenReview forum: "Expanding SPHERE-JEPA: A Family of Statistical Regularizers for the Hypersphere"
_TMLR — Under review for TMLR_

### Review · Reviewer_w9R1 · 2026-06-25

**Summary Of Contributions:**

The paper expands the regularizer of SHEPERE-JEPA by formulating full-dimensional objectives for Maximum Mean Discrepancy (MMD), Kernel Stein Discrepancy (KSD), and Kullback-Leibler divergence (KL) directly on the hypersphere to enforce a uniform distribution on the representations.

The main claims are the derivation of the hyperspherical regularization and application on the SPHERE-JEPA, and its evaluation.

The main weakness is the limited evaluation that evaluates only ResNet-19 on two datasets.  While the derivations still hold and are valid, their contribution is questionable given the lack of evidence on the current defacto architectures (transformers).

**Audience:**

No

**Audience Explanation:**

While the derivation of the regularizers is interesting, their usefulness is limited to their applicability in current models.  Given that the paper doesn't demonstrate this applicability, I'm limiting the impact to the audience of TMLR.

**Claims And Evidence:**

No

**Claims Explanation:**

The evaluation is limited to ResNet-18.  It is not clear how the proposal holds on the existing standard architectures used for representation learning and vision tasks (ViTs).

**Requested Changes:**

- Can you do your evaluation on ViTs?
- Can you show the performance on traditional dense tasks as well?

Minor concerns:
- The presentation of the main results is limited at best in Section 3.  All the details are in the appendix.  Given that there is still room on the main paper, enriching Section 3 with motivation and high-level ideas of the derivations and proofs would help the reader follow and understand the paper, and then rely on the Appendix for the proof and details of the derivation.

---

### Review · Reviewer_HbHJ · 2026-07-02

**Summary Of Contributions:**

This study addresses the gradient variance problem of sliced statistical regularization on the sphere due to Monte Carlo random projection in SSL. By analyzing the integral, the deterministic MMD form is derived, and on this basis, the precise multi-dimensional statistical test regularization family on the sphere is systematically extended. Combining spectral theory, the rotation-invariant Heat and Bandlimited kernel functions were designed, revealing the shaping effects of different statistical tests on the topological geometry of potential space.

Strengths:

1. The theoretical derivation is complete.

2. The findings in the paper have significant implications for understanding the self-supervised regularization mechanism.

3. The paper objectively points out the limitations of the $\mathcal{O}(B^2)$ quadratic computational complexity faced by exact multi-dimensional verification.

Weaknesses:

1. The scale and coverage of downstream benchmark experiments are seriously insufficient.

2. Baseline contrasts with mainstream self-supervision (specifically, contrastive learning vs. non-contrastive learning aspheric methods) are missing.

3. In the introduction section, the author introduces the motivation with the latest but not classic SPHERE-JEPA method, which is unfamiliar to readers in other subfields or even those who have long studied contrastive learning. The author should provide a slightly more detailed explanation.

In general, the research questions in this paper are meaningful and theoretically rigorous, but experimental verification is very lacking, and the TMLR Journal's focus on practical authentication theory is currently inconsistent.

**Audience:**

Yes

**Audience Explanation:**

The theme in this paper is self-supervised learning (contrastive learning), which is a classical problem in modern machine (deep) learning.

**Broader Impact Concerns:**

The potential impact of the work in this paper may be extensive, as it is fundamental research that can be applied to the fields of images, unstructured graph data, and so on.

**Claims And Evidence:**

No

**Claims Explanation:**

The theoretical analysis in this paper is complete, but the related experimental validations are incomplete, including SOTA contrastive baselines and more real-world datasets.

**Requested Changes:**

1. Experimental results need to be added to more datasets, and the datasets should be multi-domain and multi-scale.

2. The current hyperparameters are also fixed, and it is not clear whether different parameters have an impact on the conclusion.

3. Dimensional experiments are also missing, and the influence of different embedding dimensions on the three proposed methods should be evaluated.

4. The classic InfoNCE loss is equivalent to \lambda=1. The method proposed in this paper should analyze the value of \lambda.

---

### Review · Reviewer_jdrG · 2026-07-17

**Summary Of Contributions:**

This paper proposed to design Self-Supervised Learning (SSL) regularization term to force the output lying with a hyper-sphere $\mathbb{S}^{d-1}$  to avoid output collapse. The proposed method follows the sliced method approach to map the feature into 1D space and derived Maximum Mean Discrepancy (MMD), Kernel Stein Discrepancy (KSD) and KL-Divergence regularization terms. Experiments on image classification tasks and texture retrieval  are conducted to justify the effectiveness of the model.

**Audience:**

Yes

**Audience Explanation:**

The method to regularize SSL model is useful, as SSL method are widely used in pre-training of the large models.

**Claims And Evidence:**

Yes

**Claims Explanation:**

1. The analytical form of MMD, KSD and KL-divergence of the regularization are derived with detailed derivation in the Appendix.
2. The property of the proposed kernel is compared with Heat kernel and bandlimit kernel.
3. The experimental results demonstrate the effectiveness of the model.

**Requested Changes:**

1. It would be useful if the computational complexity of each proposed method is discussed and the training time is compared, as kernel based method could be time consuming.
2. It would be better if larger dataset like ImageNet 1k is tested, as the SSL model is useful in the presence of large amount of unlabeled data

---

### Comment · Action_Editor_D8sf · 2026-07-18

The paper derives deterministic, full-dimensional hyperspherical regularizers based on MMD, KSD, and KDE-based KL objectives, supported by a spectral kernel framework. Reviewers generally view the theoretical development as interesting and rigorous, but disagree on whether the current experiments sufficiently support the broader practical claims. The main concerns are limited dataset and architecture coverage, missing mainstream SSL baselines and ViT or dense-task evaluations, computational cost, hyperparameter and embedding-dimension sensitivity, and the accessibility of the presentation.

Authors, please clarify whether the primary contribution is theoretical or intended as a broadly validated practical SSL method, and prioritize the most informative additional analyses or experiments. Reviewers, please distinguish which requested experiments are essential for validating the central claims from those that are desirable extensions, and consider whether calibrated claims together with targeted evidence could adequately address the concerns.